# Synovial Gene expression after Hemarthrosis differs between FVIII-deficient mice treated with recombinant FVIII or FVIII-Fc Fusion Protein

Bilgimol Chumappumkal Joseph[1]*, Thomas C. Whisenant[2], Esther J. Cooke[1], Jenny Y. Zhou[1], Nicca Falah[1], Juan Andres De-Pablo Moreno[3], Annette von Drygalski[1]

1 Department of Medicine, Division of Hematology/Oncology, University of California San Diego, La Jolla, California, United States of America, 2 University of California San Diego, Center for Computational Biology and Bioinformatics, La Jolla, California, United States of America, 3 Department of Genetic, Physiology and Microbiology, Biology School, Complutense University of Madrid, Madrid, Spain

* bijoseph@health.ucsd.edu

## Abstract

To investigate if FVIII-Fc Fusion protein (FcFVIII) may modulate inflammation and immune stimulation in hemophilic synovium via the Fc-portion of immunoglobulin used for half-life extension we performed gene expression profiling in FVIII-deficient mice. Hemarthrosis was induced by sub-patellar puncture in FVIII-KO mice, +/- periproce-dural recombinant human (rh)FVIII, murine (m)FcFVIII, or mIgG2a. Synovium was harvested at baseline and on days (D) 3 and 14, followed by RNA extraction and sequencing, and histological analysis. RNASeq data were processed using standard protocols followed by differential gene expression (DGE) analysis. Functional enrichment analysis generated molecular pathways (KEGG and Reactome). To distinguish between on-target and off-target (related and unrelated to injury/bleed) effects the following groups were compared: i) Baseline vs. injured-saline, ii) injured-saline vs. injured-rhFVIII, iii) injured-saline vs. injured-mFcFVIII. Knee injury in FVIII-KO mice resulted in hemarthrosis, which was prevented by peri-procedural rhFVIII and mFcFVIII treatments. Only a small proportion of genes was affected by FVIII treatment, exhibiting overlap but also distinct differences between both FVIII-preparations. Acutely (D3), mFcFVIII had unique on-target effects related to immune and inflammatory regulation, whereas rhFVIII mostly affected mRNA and protein processing. On day 14, macrophage profiling indicated a transition from M1 to M2, and only mFcFVIII uniquely influenced pathways and genes associated with tissue remodeling and repair. Some mFcFVIII DGE patterns resembled mIgG2a patterns. Synovial vascular remodeling and cartilage health were better with mFcFVIII than rhFVIII. Interestingly, both FVIII-preparations exerted off-target effects on immune system pathways, albeit with temporal differences. These observations provide proof-of-principle that the type of FVIII preparation can influence synovial processes beyond acute hemostasis control, deserving exploration in the setting of joint bleed control in hemophilia.

**Data availability statement:** All relevant data are within the paper and its Supporting Information files.

**Funding:** This study was funded by the National Hemophilia Foundation/Nicholas Cirelli Family Research Fund Judith Graham Pool Research Fellowship (EJC), by a research grant from Sanofi (AvD), and UL1TR001442 of CTSA (TCW). The funders had no role in study design, data collection, and analysis. Sanofi reviewed and provided feedback on the manuscript. The authors had full editorial control of the manuscript and provided their final approval of all content.

**Competing interests:** I have read the journal's policy and the authors of this manuscript have the following competing interests: AvD has received honoraria for participating in scientific advisory board panels, consulting, and speaking engagements for BioMarin, Pfizer, Bioverativ/Sanofi, CSL-Behring, Novo Nordisk, Spark Therapeutics, Takeda and Regeneron. AvD is co-founder of Hematherix LLC., a biotech company that is developing superFVa therapy for bleeding complications. AvD is a member of the Board of Directors of Hematherix LLC. This does not alter our adherence to PLOS ONE policies.

## Introduction

Hemophilia A is an inherited bleeding disorder resulting from the deficiency of coagulation factor VIII. Patients with severe hemophilia (factor levels <1%) experience spontaneous recurrent hemarthroses [1]. Frequent blood exposure in the joint leads to synovial inflammation and vascular remodeling resulting in the development of hemophilic arthropathy (HA) [2,3]. Despite developments in clotting factor and factor-less replacement therapies [4,5], recurrent bleeding or progressive HA cannot be fully prevented. The etiology of recurrent joint bleeding, which is in part associated with vascular remodeling [6] and the ensuing development of HA, seem multifaceted and are poorly understood on a molecular level.

Transcriptomics provides a comprehensive understanding of gene expression, function, and regulation in health and disease [7], enabling investigators to reveal molecular processes in an unbiased fashion. Such non-targeted analyses have the potential to generate a comprehensive delineation of DGE and molecular pathways involved in the pathological aspects of disease, perhaps offering novel targets for diagnosis and treatment. Furthermore, secondary functional analyses that utilize the troves of previously generated data allow for estimation of cell type proportions based on established gene markers or newly generated single-cell RNA-Seq data.

Here, we used next-generation RNA-sequencing (RNA-Seq) to understand the timely evolution of molecular processes in synovial tissue after hemarthrosis in a mouse model of Hemophilia A. We investigated gene expression changes and perturbation of molecular pathways with and without Factor (F)VIII treatment 3 and 14 days after induced hemarthrosis. To elucidate if the Fc-portion of immunoglobulin, used for half-life prolongation of FVIII preparations, affects processes related to tissue inflammation, repair, or immune responses, we studied murine Fc FVIII (mFcFVIII) in comparison to recombinant human FVIII (rhFVIII, same molecule without mFc).

## Materials and methods

### Mice

FVIII-deficient mice [(FVIII-KO), BALB/c background] were provided by David Lillicrap (Queen's University, Kingston, ON, Canada) and bred at the University of California San Diego (UCSD). Housing conditions and experiments were following guidelines and protocols approved by the UCSD Institutional Animal Care and Use Committee.

### Joint injury model to induce hemarthrosis

Hemarthrosis was induced in skeletally mature (age 12–16 weeks) FVIII-KO mice (+/-rhFVIII or mFcFVIII prophylaxis by sub-patellar punctures of the right knee with a 30-gauge needle as described [6]. Mice were anesthetized with 1.5% isoflurane and 2 L/min of $O_2$ and were treated with saline (vehicle) 100 IU/kg mFcFVIII (Fc murine due to species specificity) or 120 IU/kg (to account for differences in $t_{1/2}$) rhFVIII [8,9] intravenously 2 hours before knee puncture, with a second dose 6 hours later.

Another group of mice was treated with murine immunoglobulin (IgG)2a (SCBT sc-3878; Santa Cruz Biotechnology, Dallas, TX, USA). mIgG2a was injected intravenously at the same molar concentration as mFcFVIII (mFcFVIII 100 IU/kg = mIgG2a 9.5 μg/Kg) 2 hours before knee injury with a second dose 6 hours later. This dose has previously been shown to effectively reduce joint and tail bleeding up to 24–48 hours after injection [10,11]. Injury-related bleeding into the joint was determined by a reduction in hematocrit values on day 2 post-injury as described [6,12]. Before the injury/bleed, a subcutaneous injection of extended-release buprenorphine (Buprenorphine-SR Lab 0.5–1 mg/kg) was given, a second dose of extended-release buprenorphine (Buprenorphine-SR Lab 0.5–1 mg/kg) was given 72 hours after the injury. Moist food was provided on the cage floor after the knee injury. 400 μL of sterile saline was administered subcutaneously daily for 3 days after the injury/bleed for fluid resuscitation.

## RNA extraction from synovium

In case of terminal experiments, mice were anesthetized with 1.5% isoflurane and 2 L/min of $O_2$ and euthanized by cervical dislocation. Tissue was harvested from each FVIII-KO mouse at baseline, day 3, or day 14 post-injury/bleed and placed immediately into RNAlater™ Stabilization Solution (QIAGEN, Frederick, MD, USA). RNAlater was removed before lysis in Buffer RLT. Synovial tissue was homogenized by passing through a 26G needle before passing through a QIAshredder column (QIAGEN). A QIAGEN RNeasy Plus Mini Kit was used for RNA extraction and the integrity of RNA samples was assessed using a 2100 Bioanalyzer (Agent Technologies, CA, USA).

## RNA sequencing

Total RNA (8 ng) from synovium was used to produce cDNA using a SMART-Seq® v4 Ultra® Low Input RNA Kit for Sequencing (Takara Bio, Mountain View, CA, USA). cDNA products were fragmented for 5 minutes using an S2 ultra-sonicator (Covaris, Woburn, MA, USA) with the following settings: intensity = 5; duty% = 10; burst cycles = 200; frequency sweeping mode. cDNA libraries were generated using a NEBNext® Ultra™ II DNA Library Prep Kit for Illumina (New England Biolabs, MA, USA) with 11 cycles of PCR. PCR-amplified libraries from synovial RNA were cleaned using 0.9X AmPure XP Beads (New England Biolabs, MA, USA) and sequenced on a NextSeq500 system (Illumina, San Diego, CA, USA) (75 bp; single-end) to generate approximately 20 million reads per sample. RNA sequencing was performed by the Next Generation Sequencing Core at the Institute of Genomic Medicine, UCSD, La Jolla, CA.

## RNA sequencing data analysis

Primary RNASeq data analyses were carried out by the Center for Computational Biology and Bioinformatics, UCSD. Quality control of the raw fastq files was performed using the software FastQC [13]. Gene expression values were generated for each sample in each group following alignment of reads to the mouse genome (mm10) using the STAR v2.5.3a aligner [14], estimation of gene-level counts using RSEM v1.3.0 [15], and GENCODE annotation (genocode.v19.annotation.gtf). BioConductor packages edgeR [16] and Limma [17] implemented within the R statistical computing environment (www.cran.org) were used to implement trimmed mean of M-values (TMM) normalization [18] and the limma-voom method [17] for differential expression analysis at both the gene and transcript levels. Lowly expressed genes were filtered out (counts per million >1 in ≥10 samples) and the experimental design was modeled upon the condition and controlled for batch effect (~0 + condition +batch). The lmFit and eBayes functions in limma were used to fit the design on voom-normalized counts per gene. Significance was defined by an adjusted p-value <0.05 after multiple testing corrections using a moderated t-statistic in limma [19]. Functional enrichment analysis was performed using the STRING database of predicted protein-protein interactions [20] or the R package gprofiler [21]. Molecular pathways from the Kyoto Encyclopedia of Genes and Genomes (KEGG) and Reactome ontology databases were retained. S1 Fig depicts timelines and processing of RNA sequencing.

## Gene expression analysis after hemarthrosis

Gene expression profiles related to FVIII treatments were defined as "on-target" (related to injury/bleed), in contrast to "off-target" (unrelated to injury/bleed) effects. Both, "on-target" and "off-target" genes are differentially expressed between injured-saline vs. injured-rhFVIII/mFcFVIII treatments; however, only "on-target" genes are differentially expressed between injured-saline vs. baseline. Synovial gene expression was compared as follows on days 3 and 14: i) "Baseline" (uninjured FVIII-KO mice) vs. FVIII-KO injured mice treated with saline ("injured-saline"); ii) injured-saline vs. FVIII-KO injured mice treated with rhFVIII ("injured-rhFVIII"); iii) injured-saline vs. FVIII-KO injured mice treated with mFcFVIII ("injured-mFcFVIII") (n = 3–5 mice per group, S1 Fig). DGE changes related to injury/bleed were identified by comparing baseline to injured-saline mice. To elucidate the treatment effects of each FVIII-preparation, injured-saline were compared to injured-FVIII treated mice (injured-rhFVIII or injured-mFcFVIII on days 3 and 14).

## Network and enrichment analysis

To understand the context of the list of differentially expressed genes (DEGs) we used them as input into the STRING interaction database [20]. This produces a network of nodes and edges that can be analyzed for enriched pathways and functions within KEGG pathways, and Reactome pathways with adjusted p-value (<0.05). Further visualization of this network was performed in Cytoscape [22].

## Cell-type-specific gene expression

Estimation of proportions for specific cell types in each sample was performed using the CIBERSORTx webtool (https://cibersortx.stanford.edu) [23]. Briefly, the algorithm creates a support vector machine model for predicting the percentage of cell types in an unknown sample using a predefined table of signature gene expression values for each cell type. In this analysis, the model was trained with the ImmGen mouse expression dataset (https://www.immgen.org) [24] which primarily comprises immunological cell types and generalized cell types including endothelial, epithelial, and stem cells. TMM-normalized Counts Per Million (CPM) for each signature gene in each synovial tissue RNA-Seq sample were analyzed with this trained model and the predicted cell percentages were averaged by time point and treatment for visualization.

## Joint tissue processing and histological analysis

Joint tissue harvested from each FVIII-KO mouse at baseline, day 3, or day 14 post-injury/bleed was fixed in 10% zinc-buffered formalin (Z-Fix) (Anatech, Battle Creek, MI) and processed as described previously [25]. In brief, the tissues were fixed and decalcified in 10% (v/v) formic acid +/-0.2% (w/v) potassium ferrocyanide for Perls' Prussian blue staining of ferric iron ($Fe^{3+}$). Decalcified joints were processed, embedded in paraffin wax, and sectioned (4µm).

Gross morphology, synovial hyperplasia, cartilage erosion, and hemosiderin were visualized by Safranin-O-Green staining as previously described [6,12]. (Semi)quantitative scoring was performed using the modified Valentino scoring system (without vascularity assessment) [26], assessing synovial hyperplasia, hemosiderin, blood in the joint, villus formation, and cartilage erosion. Vascularity assessment was performed separately with α-smooth muscle actin (αSMA) as a marker for remodeling vessels using rabbit polyclonal anti-αSMA (ab5694; Abcam, Cambridge, UK) as previously described [25]. Glycosaminoglycan (GAG) intensity (tibia and femur cartilage surfaces) was determined by Safranin-O-Green staining [6,27] and quantified using ImageJ software (National Institutes of Health, Bethesda, MD, USA) to determine GAG loss. Slides were imaged using a NanoZoomer 2.0-HT brightfield slide scanner at 20x magnification (Hamamatsu Photonics, Hamamatsu, Japan).

## Statistical analyses

Data are expressed as median plus interquartile range. Statistical significance was determined by Mann-Whitney test (when comparing two groups) or Kruskal Wallis with Dunn's multiple-comparison test (when comparing more than two groups). A p-value <0.05 was considered statistically significant. For RNA-Seq data, the criteria for DGE was an adjusted p-value <0.05. Pathways with an over-representation adjusted p-value <0.05 were considered significantly enriched.

## Results

### Magnitude of differential gene expression changes after hemarthrosis in FVIII-KO mice

Hemarthrosis by needle puncture is used frequently to study joint bleeding tendencies, treatment efficacy, and pathobiology of hemophilic arthropathy [6,12]. The severity of joint bleeding can be assessed objectively by observing a drop in mean hematocrit. Here, needle puncture resulted in severe joint bleeding (drop in mean hematocrit from 47.5% to 25% on day 2 in mice receiving saline and mIgG2a), rescued by peri-procedural treatment with both FVIII preparations (mean day 2 hematocrit rhFVIII: 45.5%; mFcFVIII: 46%) (S2 Fig).

Table 1 displays the number of DEGs for each treatment and time point based on the groupings listed in method section. The comparison of baseline vs injured-saline mice yielded 3305 DEGs on day 3, and this response was reduced by 58% with rhFVIII and by 74% with mFcFVIII. On day 14, the number of DEGs for injured-saline mice was much lower than on day 3 (1606 DEGs) and reduced to 94% by rhFVIII and 86% by mFcFVIII. Overall, it appeared that the extent of DGE suppression was roughly comparable between mFcFVIII and rhFVIII on both days. A complete list of DEGs, with their direction of change, is provided in S1 Table.

### DEGs 3 days after hemarthrosis

Overlapping and unique gene expression profiles for each comparison, listed in Table 1, were converted into a Venn diagram to delineate gene expression profiles related to FVIII treatments as "on-target" (related to injury/bleed), "off-target" (unrelated to injury/bleed), and unaffected. Treatment with FVIII preparations resulted in gene expression changes affecting similar numbers of on-target (n = 956) and off-target (n = 652) genes, while the majority of genes remained unaffected by FVIII treatments (n = 2349) (Fig 1A). Gene expression changes fell into seven distinct groupings (Figs 1B-1D), separating on- and off-target changes in relation to FVIII treatments. Among 3957 DEGs, on-target gene expression unique to rhFVIII (group 1) and mFcVIII (group 2) was identified for 387 (9.7%) and 106 genes (2.6%), respectively

**Table 1. Differentially expressed genes in the synovial tissue from FVIII-KO mice after hemarthrosis and FVIII treatment.** Synovial tissue was harvested at baseline, day 3 and day 14 post-injury. RNA was purified and analyzed by RNA sequencing using an Illumina NextSeq500 platform (75 bp; single-end). The R BioConductor packages tximport, edgeR, and limma were used to read estimate counts from RSEM, trimmed mean of M-values (TMM) normalization was applied, and the limma-voom method was used for differential expression analyses (criteria: adjusted p-value <0.05). The baseline group was compared to saline treatment groups, and saline-treated groups were compared to rhFVIII and mFcFVIII treatment groups on day 3 and day 14. This table lists the total number of DEGs and the number of up or down-regulated DEGs in each group.

| Comparison | Post-injury days | Total number of DEGS | Number of DEGs up-regulated | Number of DEGs down-regulated |
|---|---|---|---|---|
| Baseline vs. Injured-Saline | Day 3 | 3305 | 1646 | 1659 |
| Injured rhFVIII vs. Injured-Saline | Day 3 | 1374 | 603 | 771 |
| Injured mFcFVIII vs. Injured-Saline | Day 3 | 873 | 328 | 545 |
| Baseline vs. Injured-Saline | Day 14 | 1606 | 659 | 947 |
| Injured rhFVIII vs. Injured-Saline | Day 14 | 90 | 37 | 53 |
| Injured mFcFVIII vs. Injured-Saline | Day 14 | 229 | 119 | 110 |

DEGs, differentially expressed genes; rhFVIII, recombinant human factor VIII; mouse-specific Fc fusion factor VIII.

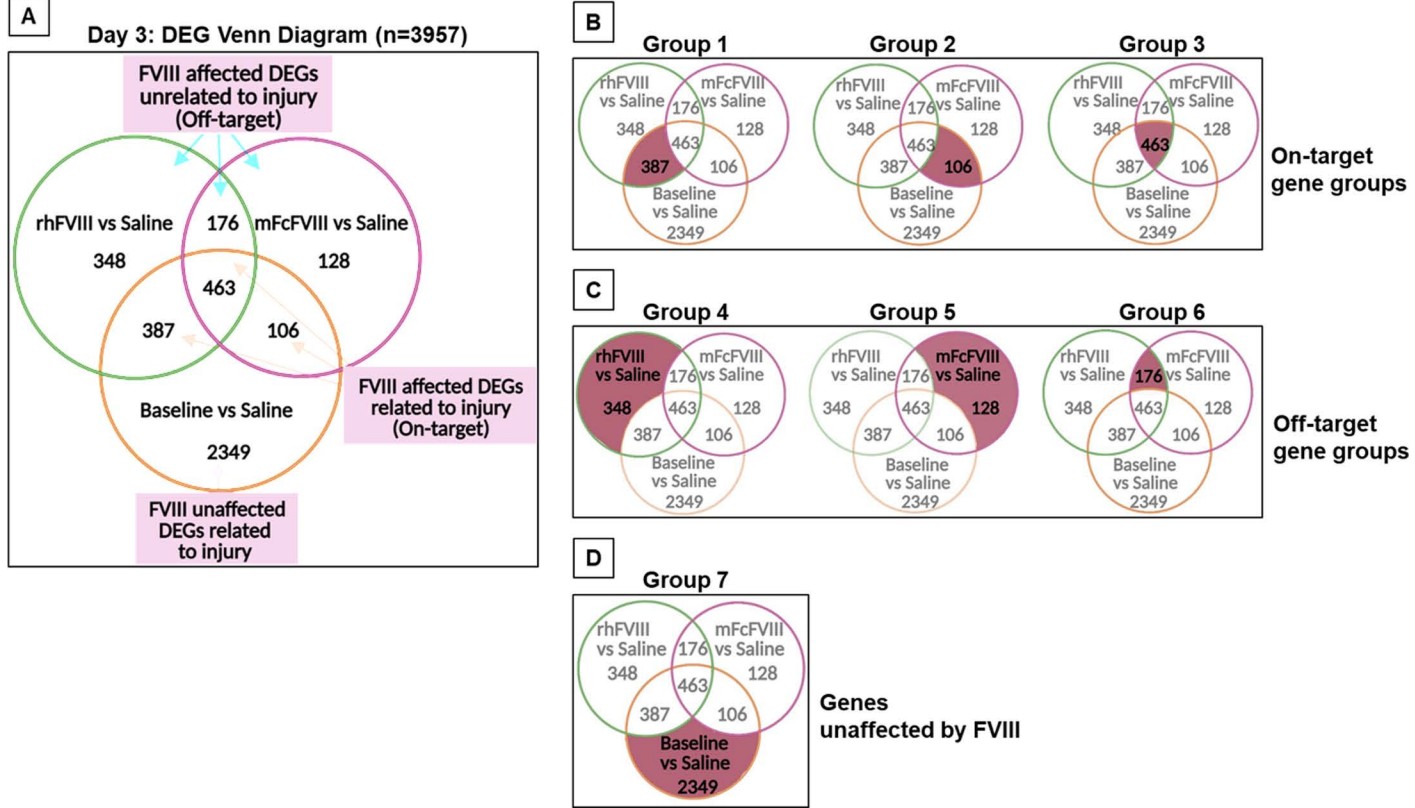

**Fig 1. Distribution of gene expression among treatment groups on day 3.** Synovial tissue was harvested at baseline and 3 days post-injury/bleed. Differential gene expression was compared as follows: baseline vs injured-saline, injured-rhFVIII vs injured-saline, or injured-mFcFVIII vs injured-saline. A Venn diagram was generated to demonstrate overlap and unique expression of DEGs in each comparison (A). Genes related to injury/bleed and affected by FVIII treatments were termed "on-target" genes (group 1: uniquely influenced by rhFVIII; group 2: uniquely influenced by mFcFVIII; group 3: influenced by both FVIII preparations) (B) Genes unrelated to injury/bleed and affected by FVIII treatment were termed "off-target" genes (group 4: uniquely influenced by rhFVIII; group 5: uniquely influenced by mFcFVIII; group 6: influenced by both FVIII preparations) (C). Genes unaffected by both FVIII treatments (group 7) (D). DEGs, differentially expressed genes; rhFVIII, recombinant human factor VIII; mFcFVIII, mouse-specific Fc fusion factor VIII.

(Fig 1B). For off-target effects (unrelated to injury/bleed), 348 DEGs (8.7%) were unique to rhFVIII (group 4), and 128 (3.2%) to mFcFVIII (group 5) (Fig 1C). The majority of gene expression changes were unaffected by either FVIII preparation (group 7: 2349 of 3957 DEGs; 59%) (Fig 1D). These findings showed that peri-procedural FVIII treatments affected only a very small proportion of gene expression changes but with unique expression profiles attributable to each of the FVIII preparations.

To determine to what extent DGE in the biological pathways overlapped or differed between the seven groups (Fig 1), we performed enrichment analysis using gprofiler to identify significantly enriched functional pathways from the KEGG (pathways denoted by "mmu") and Reactome (pathways denoted by MMU-), summarized in S2 Table. Group 1 (on-target, unique to rhFVIII) incited multiple pathways that were related to RNA processing and protein production (MMU-8953854, MMU-392499). In contrast, group 2 (on-target, unique to mFcFVIII), incited the cytokine-cytokine receptor interaction (mmu04060) pathway, which includes TNF receptor superfamily genes, chemokine ligands, chemokine receptors, and Interleukin (Il)1b Group 3 (on-target, both FVIII preparations) resulted in multiple enriched pathways with overlapping elements from each of the previously mentioned groups. For group 4 (off-target, unique to rhFVIII), perturbation of immune

system (MMU-168256) and metabolism (MMU-1430728) pathways were identified, whereas no pathways were perturbed in group 5 (off-target, unique to mFcFVIII). Group 6 (off-target, both FVIII preparations) was enriched for changes in the metabolism (MMU-1430728) and cell adhesion (mmu04514) pathways. Group 7 (unaffected by FVIII preparations) contained a large number of significant pathways, comprising mostly extracellular matrix (ECM) organization (MMU-1474244) and cell cycle (MMU-1640170) related genes. The genes contributing to the pathways in the 7 groups (S2 Table) are visualized as networks based on their interactions in the STRING database [20] (S3A-S3G Figs). These observations indicate that both FVIII preparations (as per DGE in groups 2, 3, and 6) affected immune and inflammatory pathways, albeit in different ways.

To explore the effects of the two FVIII preparations further, we sought to identify the genes whose on-target expression changes were reversed with FVIII, and to what extent immune regulation was involved. Therefore, we analyzed groups 1 and 2 (Fig 1) by plotting the log-fold gene expression changes for the comparisons of baseline vs injured-saline against injured-rhFVIII or injured-mFcFVIII vs injured-saline (Fig 2A). Almost all genes showed a reversal of injury/bleed-induced expression changes with both FVIII preparations. Reversal is defined as increased DGE (log fold change [LogFC]) in the injured-saline group, which was dampened with either FVIII preparation or vice versa.

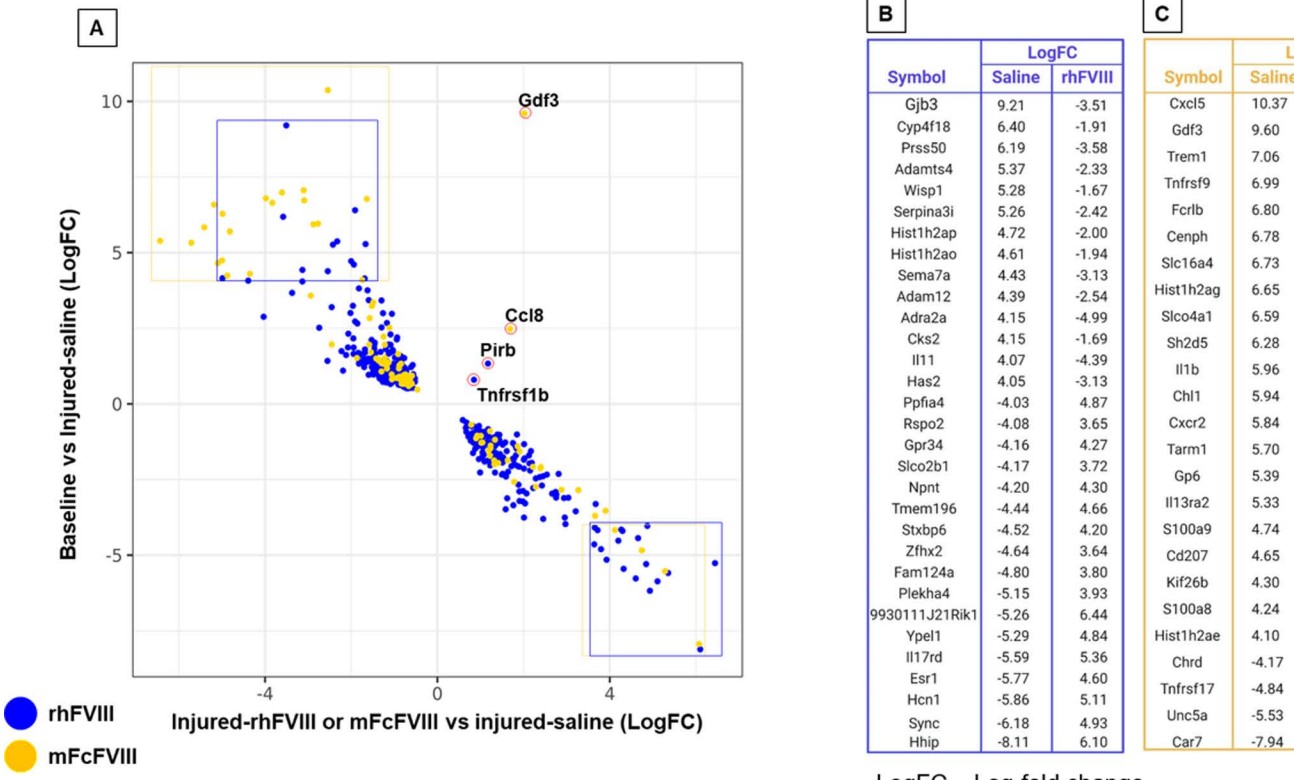

**Fig 2. Directionality of synovial gene expression +/−FVIII treatment 3 days after injury/bleed.** DEGs of groups 1 (unique to rhFVIII – blue) and 2 (unique to mFcFVIII – gold) were selected and a scatter plot was generated with LogFC gene expression values (A). B and C lists all genes from groups 1 and 2 with LogFC values >4.0 or <-4.0 following saline treatment compared to baseline and gene expression changes achieved with rhFVIIII and mFcFVIII, respectively. DEGs, differentially expressed genes; rhFVIII, recombinant human factor VIII; mFcFVIII, mouse Fc recombinant factor VIII; LogFC, Log fold change.

Since all DEGs unique to groups 1 and 2 were reversed with rhFVIII or mFcVIII treatment, we focused specifically on the genes that had the highest differential expression of LogFC ≥ 4.0 or ≤-4.0. We found that the majority of genes were up-regulated (LogFC ≥ 4.0) with injury/bleed and that most of the reversed genes belonged to group 2. Most of the genes down-regulated (LogFC ≤ -4.0) with injury/bleed and reversed belonged to group 1 (Fig 2A). Figs 2B-2C lists the genes unique to reversal with rhFVIII (group 1) and mFcFVIII (group 2), revealing that 9 of 25 genes unique to mFcFVIII have biological relevance in immune-modulatory and inflammatory processes, whereas genes unique to rhFVIII involved mostly cell proliferation, adhesion, and signaling. Of note, there were four genes (all macrophage markers) that were upregulated with injury/bleed and further augmented by FVIII treatment (rhFVIII: *Pirb*, *Tnfrsf1b*; mFcFVIII: *Gdf3*, *Ccl8)*. Gene expression patterns were also visualized side-by-side using heatmaps, providing visual characterization of the uniquely up- and/or downregulated genes. (S4A and S4B Figs). Together, these findings demonstrate unique effects of rhFVIII and mFcFVIII on synovial gene expression early after injury/bleed, particularly in relation to genes involved in immune regulation and inflammation.

## DEGs 14 days after hemarthrosis

In analogy to Day 3, overlapping and diverging gene expression profiles for each comparison (Table 1) are depicted in a Venn diagram, enabling the identification of "on-target" and "off-target" genes affected and unaffected by rhFVIII or mFcFVIII (Fig 3A). Overall, gene expression changes were far less on day 14 compared to day 3, but rhFVIII and mFcFVIII still elicited unique on-target and off-target gene expression patterns. As for day 3, gene expression changes were categorized into seven distinct groupings (Figs 3B-3D). Among 1728 on-target DEGs, those unique to rhFVIII (group 1) and mFcVIII (group 2) were 20 (1.2%) and 95 genes (5.5%), respectively (Fig 3B). Off-target, 28 DEGs (1.6%) were unique to rhFVIII (group 4), and 92 (5.3%) to mFcFVIII (group 5) (Fig 3C). Therefore, it appeared that rhFVIII resulted in fewer gene expression changes compared to mFcFVIII. However, most gene expression changes were unaffected by either FVIII preparation (group 7: 1451 of 1728 DEGs; 84%) (Fig 3D).

Analysis using gprofiler to identify significantly enriched functional biological pathways from the KEGG and Reactome databases was performed for the seven groups (Figs 3B-3D). Group 1 had zero enriched pathways, while groups 2 and 3 were significantly enriched for the ECM organization (MMU-1474244) and the collagen degradation (MMU-1442490) pathways, with some genes also contributing to the focal adhesion (mmu04510) pathway in group 3 (S5A and S5B Figs). These findings suggest that mFcFVIII exerted unique on-target effects on pathways involved in tissue reorganization. For group 4 (off-target, unique to rhFVIII) and group 5 (off-target, unique to mFcFVIII), the immune system pathway (MMU-168256) was significantly enriched, with different DEG sets contributing, while group 6 had zero enriched pathways. Together this suggests unique off-target immune effects for each FVIII-preparation. Group 7 contained a large number of enriched pathways that were mostly related to ECM organization and metabolism (MMU-1430728) (S5C-S5F Figs). A full list of genes and pathways is provided in S2 Table.

In analogy to day 3, we explored FVIII treatment effects on day 14 by comparing the log-fold gene expression changes of baseline vs injured-saline against injured-rhFVIII or injured-mFcFVIII vs injured-saline. The majority of injury/bleed-related genes showed a reversal with both FVIII treatments, similar to day 3. When focusing specifically on the genes with the highest differential expression (LogFC ≥ 4.0 or ≤ -4.0) after injury/bleed, three DEGs were downregulated and reversed by rhFVIII, and nine DEGs were reversed by mFcFVIII (seven upregulated; two downregulated) (Figs 4A-4C). Most of the mFcFVIII affected genes (*Tnn*, *Col21a1*, *P4ha3*, *Grid2,* and *Timd4)* contributed to tissue repair/remodel functions, also visualized using heat maps (S6A and S6B Figs).

Altogether, the observed gene expression changes after injury/bleed suggested a transition from a predominantly inflammatory state on day 3 to tissue remodeling on day 14, accompanied by fluctuating on- and off-target gene expression changes related to immune modulation. Since many changes were affected uniquely by rhFVIII or mFcFVIII we suspected FVIII-specific cellular flux and/or proliferation, prompting cell-type deconvolution analysis.

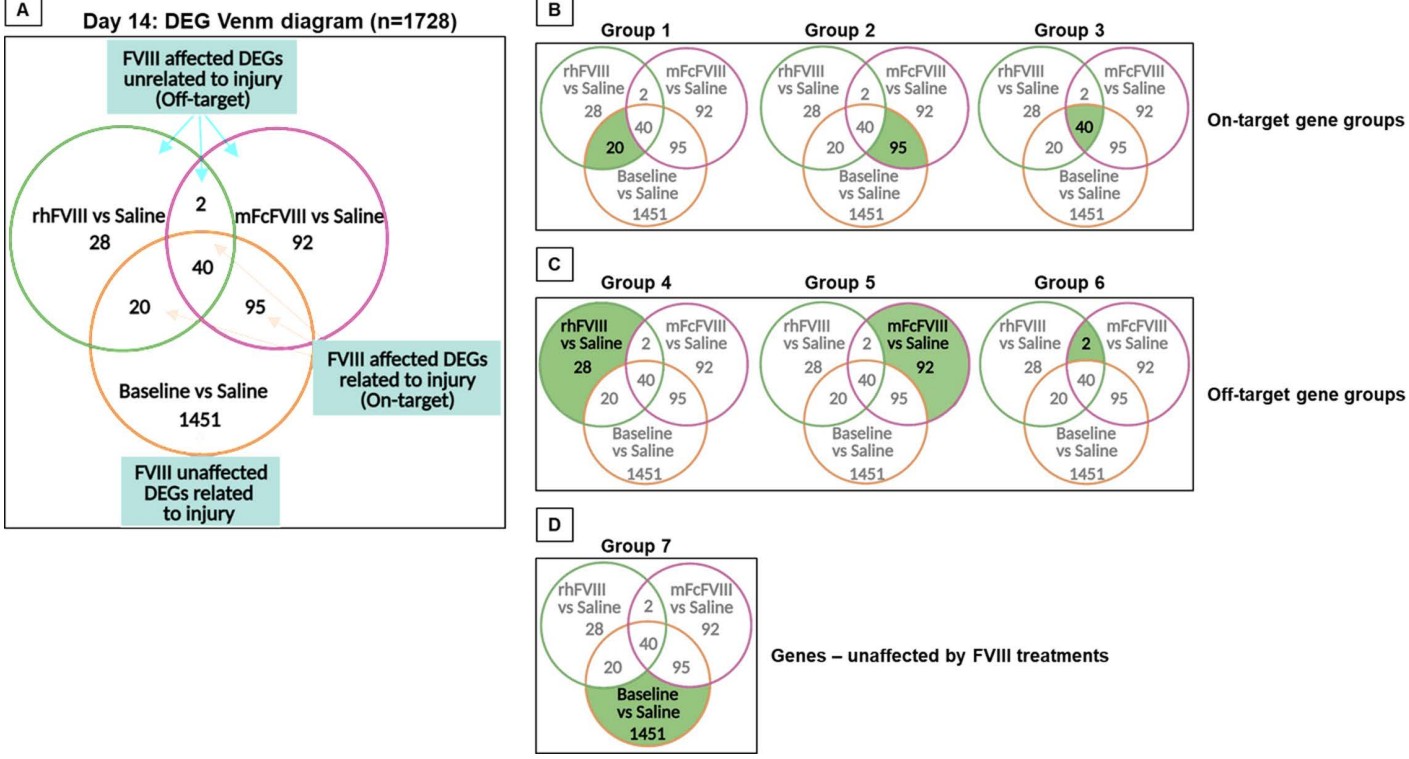

**Fig 3. Distribution of gene expression among treatment groups on day 14.** Hemarthrosis was induced in FVIII-KO mice by sub-patellar needle puncture. Mice were treated with saline, rhFVIII or mFcFVIII 2 h before and 6 h after injury (n = 3 to 5 per group). Synovial tissue was harvested at baseline and 14 days post-injury. Differential gene expression was compared as follows: baseline vs injured-saline, injured-rhFVIII vs injured-saline, or injured-saline vs injured-mFcFVIII. A Venn diagram was generated to demonstrate overlap and unique expression of DEGs in each comparison (A). Genes related to injury/bleed and affected by both FVIII treatments were termed "on-target" genes (group 1: uniquely influenced by rhFVIII; group 2: uniquely influenced by mFcFVIII; group 3: influenced by both FVIII preparations) (C). Genes unrelated to injury/bleed and affected by FVIII treatment were termed "off-target" genes (group 4: uniquely influenced by rhFVIII; group 5: uniquely influenced by mFcFVIII; group 6: influenced by both FVIII preparations (B). Genes unaffected by both FVIII treatments (group 7) (D). DEGs, differentially expressed genes; rhFVIII, recombinant human factor VIII; mouse-specific Fc fusion factor VIII.

## Cell-type deconvolution from bulk RNASeq using immune cell references

We previously showed pronounced synovial proliferative tissue expansion after induced joint bleeding in FVIII-KO mice with strong expression of the proliferating cell nuclear antigen [6]. Here, we used deconvolution analysis to infer the presence and proportion of different cell types based on treatment-specific gene expression profiles. We analyzed the immune-related cell types during the acute/early phase (day 3) and the transition phase from M1 to M2 polarization after hemarthrosis (day 14) [12], representing the transition from inflammatory to reparative processes.

Briefly, the IMMGEN dataset [28] was used to generate a custom signature matrix for imputing cell fractions using the CIBERSORTx tool [23]. The most prevalent cell populations were macrophages and fibroblasts, comprising ~80% of all cell types, consistent with reported synovial cell compositions [6]. The remainder included small proportions of epithelial, natural killer, stem, and T- cells. Compared to saline treatment after hemarthrosis, both FVIII preparations resulted in a comparable mild decrease of fibroblast proportions during acute phase (day 3) and transition phase (day 14). On day 3 both rhFVIII and mFcFVIII demonstrated an increase in the macrophage population (Fig 5A), which persisted with mFcFVIII on day 14, while it returned to baseline levels with rhFVIII. (Fig 5B).

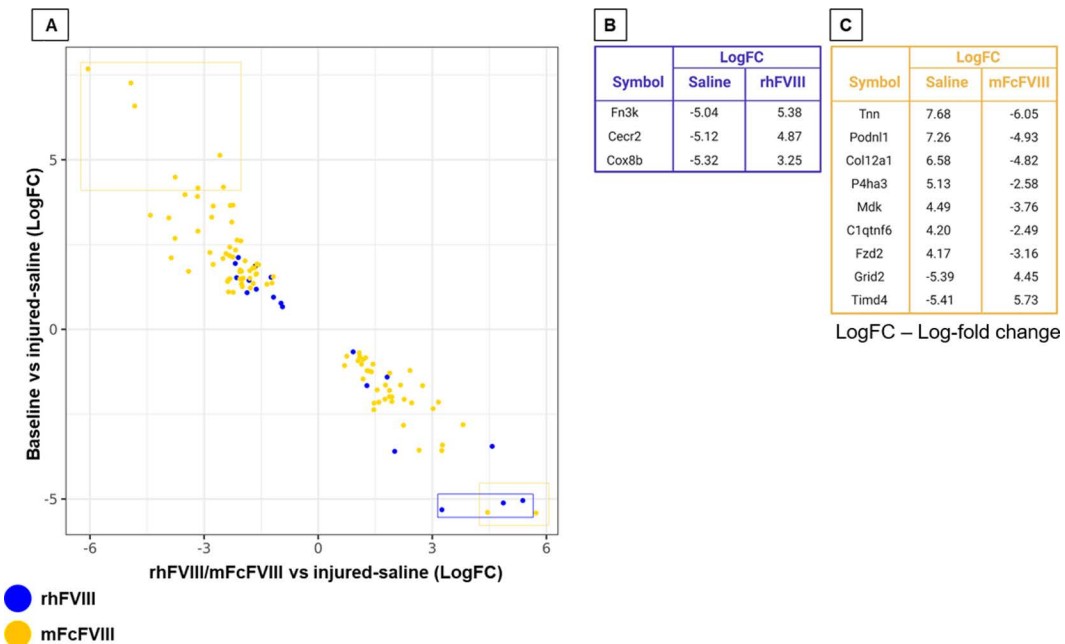

**Fig 4. Directionality of synovial gene expression +/−FVIII treatment 14 days after injury/bleed.** DEGs of groups 1 (unique to rhFVIII – blue) and 2 (unique to mFcFVIII – gold) were selected and a scatter plot was generated with LogFC gene expression values (A). Table A and B lists all genes from groups 1 and 2 with LogFC values >4.0 or <-4.0 following saline treatment compared to baseline and gene expression changes achieved with rhFVIII and mFcFVIII, respectively. DEGs, differentially expressed genes; rhFVIII, recombinant human factor VIII; mouse-specific Fc fusion factor VIII; LogFC, Log fold change.

To this end, we aggregated the expression profiles of common M1 and M2 sub-type markers (M1: *Il6*, *Il1β*, *Nos2*, *Cxcl2* and *Il1r1*; M2: *Arg1*, *Chil3*, *Mmp9*, *Fcgr4*, *Mgl2* (*Clec10a*), *Mrc1*, *Pparg*, and *Irf4*) [12]. During the acute phase (day 3), all M1 markers were elevated in response to hemarthrosis (*Il6*, *Il1β*, *Nos2*, *Cxcl2* and *Il1r1*), whereby *Il6* and *Nos2* were suppressed to baseline levels by rhFVIII and mFcFVIII alike, without effects on *Il1β*, *Cxcl2* and *Il1r1* (Fig 5C). However, during the transition period (day 14) only *Il6* remained mildly increased in response to hemarthrosis, and was somewhat suppressed with both FVIII-preparations, indicating lesser inflammation on day 14 (Fig 5D). Pertaining to M2 markers, six of eight M2 markers were affected (up-/down-regulated with hemarthrosis and/or affected by FVIII-treatments) on day 3 and 14 (*Arg1*, *Chil3*, *Mmp9*, *Fcgr4*, *Pparg*, *Irf4*). On day 3 all six M2 markers were either up- or down-regulated after hemarthrosis, and both FVIII preparations corrected five of six markers towards baseline (exception *Fcgr4*) (Fig 5E). On day 14, three of the six markers remained altered in response to hemarthrosis (*Mmp9*, *Fcgr4*, *Pparg*) without correction by FVIII, with the exception of *Mmp9*, which was normalized with mFcFVIII only. Among the three M2 markers unaltered by hemarthrosis (*Arg1*, *Chil3*, *Irf4*), *Arg1* and *Chil3* demonstrated a notable decrease and increase with rhFVIII, respectively, while relatively unaffected (similar to baseline) with mFcFVIII (Fig 5F). Together, these findings indicate a change in differential expression profiles of M1/M2 markers during the healing phase, influenced to some extent by the FVIII preparations, with some notable differences between rhFVIII and mFcFVIII. However, the biological significance of those markers, particularly inflammatory or anti-inflammatory functions, remains unknown in this context.

### Synovial and cartilage changes after hemarthrosis

Since RNASeq data indicated a shift from inflammatory to reparative processes on day 14; we conducted a histological examination of joint health during this transitional phase. Joints were scored for synovial and cartilage changes (day 14) and compared to baseline knee joints. Hemarthrosis resulted in pronounced synovial hyperplasia, increased hemosiderin

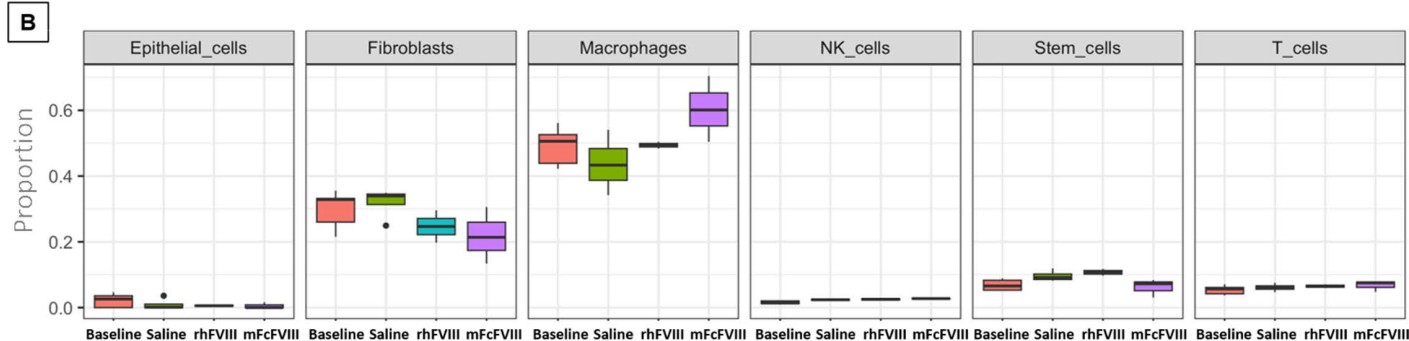

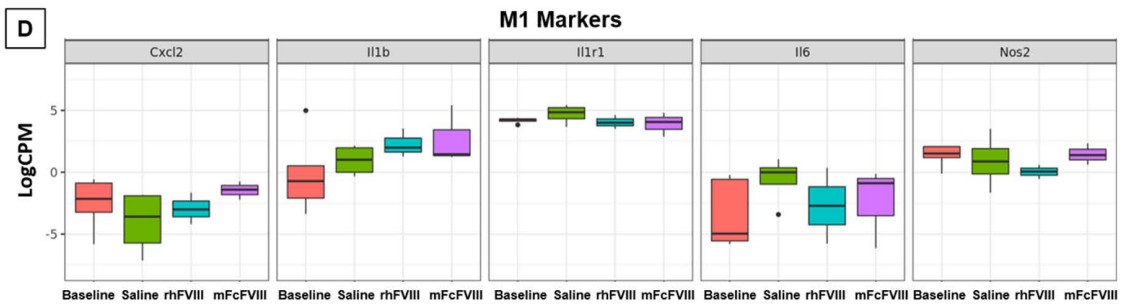

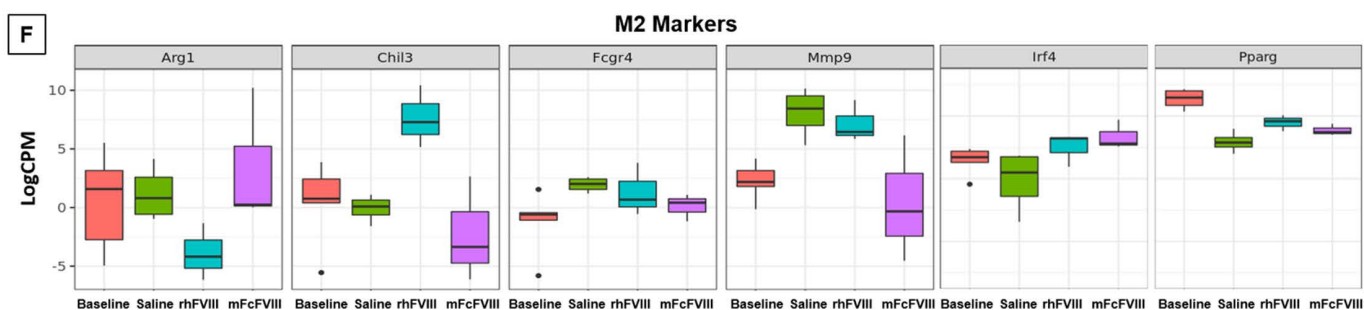

**Fig 5. Cell type-specific gene expression in synovium on days 3 and 14.** In order to identify the cell types, deconvolution analysis was used to estimate the proportions of a range of immune cells. Briefly, the IMMGEN dataset was used to generate a custom signature matrix for imputing cell fractions using the CIBERSORTx tool. Different cell populations were identified on days 3 and 14 (A and B). Differential expression levels of macrophage subtype-specific markers M1 (C and D) and M2 were determined by LogFC values (E and F). IMMGEN, Immunological genome project; LogFC, log fold change; rhFVIII, recombinant human factor VIII; mFcFVIII, mouse-specific Fc fusion factor VIII; NK cells, natural killer cells; Il6, interleukin 6; Il1b, interleukin 1b; Nos2, nitric oxide synthase 2; Cxcl2, C-X-C motif chemokine ligand 2; Ilr1, Interleukin 1 receptor type 1; Fcgr4, Fc gamma receptor 4; Irf4, Interferon regulatory factor 4; Pparg, peroxisome proliferator activated receptor gamma; Arg1, arginase-1; Chil3, chitinase-like protein 3; and Mmp9, matrix metalloproteinase 9.

deposition, cartilage erosion, and blood in the joint (median Valentino score 5; range 3–6). Scores improved significantly with both FVIII preparations (rhFVIII: median 3, range 1–3; p=0.0122; mFcFVIII: median 2, range 1–4; p=0.002) (Fig 6A). Vascular density increased significantly after hemarthrosis (p=0.0007), remained unchanged with rhFVIII, but decreased significantly with mFcFVIII (p=0.008) (Fig 6B). Similarly, cartilage GAG content decreased significantly after hemarthrosis (p=0.004), recovered to baseline with rhFVIII, but augmented significantly above baseline with mFcFVIII (p=0.002; Fig 6C). To that end, only 3/8 (37.5%) mice treated with mFcFVIII compared to 5/6 (83.3%) mice treated with rhFVIII had cartilage erosions assessed via the Valentino scoring algorithm [26]. Representative examples are depicted in Fig 6D. These results suggest that mFcFVIII decreases synovial vascular remodeling, and promotes cartilage health in a meaningful way.

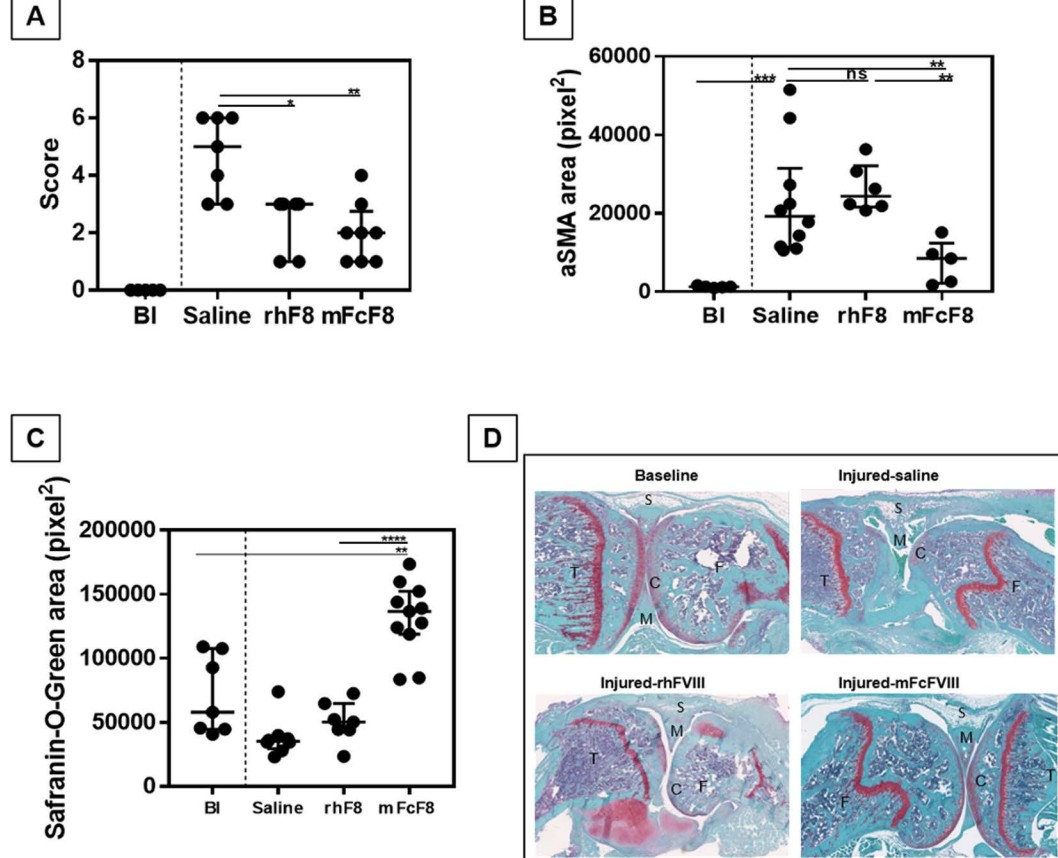

**Fig 6. Soft tissue and cartilage changes following induced hemarthrosis in FVIII-KO mice.** The right knee of FVIII-KO mice was subjected to subpatellar puncture. Joint tissue was harvested on day 14 after hemarthrosis (n = 5-8 per group). (A) Soft tissue damage was studied by histopathology using Valentino scoring (without vascularity assessment) after Safranin-O-Green staining (maximum score = 7). (B) Immunohistochemistry was performed with an anti-αSMA antibody to detect vascular changes. (C) Cartilage health was assessed by ImageJ analysis of the cartilage after Safranin-O-Green staining. (D) Representative examples of baseline (Bl) and injured-saline (saline), injured-rhFVIII (rhF8), and injured-mFcFVIII (mFcF8) are displayed. αSMA, alpha smooth muscle actin; Bl, baseline; rhF8/rhFVIII, recombinant human factor VIII; mFcF8/mFcFVIII, mouse-specific Fc fusion factor VIII; S, synovium; M, meniscus; T, tibia; F, femur; C, cartilage.

## Effects of mouse IgG2a on hemarthrosis

Murine IgG2a was selected as a control agent to determine independent effects of the Fc-portion of mFcFVIII because rhFVIII is fused to murine IgG2a-Fc (mFcFVIII) due to species specificity as previously reported [29]. We hypothesized that the number of DEGs between the injured-IgG2a vs injured-rhFVIII groups would be greater than between the injured-mIgG2a vs injured-mFcFVIII groups. To this end, the number of DEGs on days 3 and 14 was indeed both higher in the injured-mIgG2a vs injured-rhFVIII compared to the injured-mIgG2a vs injured-mFcFVIII groups. On day 3 there were 1335 DEGs (injured-mIgG2a vs injured-rhFVIII) and 964 DEGs (injured-mIgG2a vs injured-mFcFVIII); moreover, on day 14, there were 28 DEGs and 0 DEGs, respectively. This indicates that the Fc-portion of the IgG influences gene expression patterns after hemarthrosis. When applying network and associated enrichment analysis to the 28 DEGs emerging in the injured-mIgG2a vs injured-rhFVIII analysis on day 14 (transition to healing phase), we found that they contributed to the Immune System pathway (MMU-168256) and the Neutrophil Degranulation Pathway (MMU-6798695). Since 0 DEGs were noted between injured-mIgG2a vs injured-mFcFVIII one may conclude that the presence of the Fc-portion abrogated

differential gene expression related to these pathways. All details about the DEGs (days 3 and 14) (S3 Table) and the pathways (S4 Table) are given in the supporting information.

## Discussion

The understanding of synovial molecular events following hemarthrosis may advance management algorithms to mitigate the development of HA. Current guidelines recommend access to prophylactic and acute treatment with clotting factor concentrates and other hemostatic agents [30]. Several FVIII preparations are available for the treatment of Hemophilia A, and they differ in respect to B-domain deletion, residual/molecular composition, and glycosylation patterns, influencing properties such as immunogenicity, metabolism, and functionality [31–34]. In addition, some FVIII products are half-life extended by adding molecules such as polyethylene glycol or Fc domain of IgG (Fc fusion). While all FVIII preparations are efficacious to replace missing congenital FVIII activity in plasma, it is unknown if they differ regarding direct effects on synovial health. It has been described that the use of FVIII Fc fusion protein improves joint health and pain [35,36], which may be independent of half-life extension and favorable pharmacokinetic parameters, particularly in respect to synovial vascular remodeling and leakiness [37]. Therefore, direct effects of the Fc component should be considered given that its anti-inflammatory and immunomodulatory properties are implied in the mechanism of action of intravenous immunoglobulin (IVIG) [38]. To this end, there is evidence that the FVIII Fc fusion protein enhances M2 macrophage polarization, mitigating immunogenicity of exogenously administered clotting factor products in mouse models [39,40]. Furthermore, rapid, and efficacious immune tolerization in individuals with hemophilia A who developed inhibitory antibodies against FVIII has been described recently [41].

Here we applied RNASeq, complimented by KEGG and Reactome pathway analysis to dissect effects of FcFVIII in comparison to FVIII on synovial molecular processes in the context of hemarthrosis. Due to species specificity of Fc-receptor interactions the human Fc component in rhFVIII (Eloctate®, Sanofi, Bridgewater, NJ, USA), was replaced by mFc. With exception of the added Fc molecule, the molecular composition of both FVIII products was identical, heightening assurance that any observed differences would be linked to the effects of Fc.

Notwithstanding small amounts of blood due to needle injury, both FVIII preparations effectively prevented hemarthrosis. However, both preparations showed unique on-target (related to injury/bleed) and off-target (unrelated to injury/bleed) effects on DGE, although the majority of synovial DEGs were unaffected by either FVIII preparation or FVIII effects overlapped. Unique effects, specific to either rhFVIII or mFcFVIII, were detected during the acute phase (day 3) as well as at the time of healing (day 14). Differences were made apparent through synovial gene expression profiling and molecular pathway analyses. In respect to pathway perturbances, differences were inherent to the incitement of distinct pathways involved in the regulation of inflammation and the immune system, as well as tissue remodeling. Acute, unique on-target effects of mFcFVIII were related to the immune system and regulation of inflammation, in contrast to rhFVIII, which uniquely affected RNA processing and protein production. Moreover, detailed gene expression analyses revealed that ~1/3 of highly up-regulated genes were immune/inflammatory regulators, all reversed with mFcFVIII. In contrast, most of highly down-regulated genes were involved in cell proliferation, adhesion, and signaling, all reversed with rhFVIII. These differences suggest a pivot from predominantly proliferative regulation early after hemarthrosis with FVIII treatment to immune and inflammatory modulation when an Fc molecule is added.

On day 14, molecular profiling revealed a stark decrease in DGE activity compared to day 3, and expression changes indicated a transition to synovial tissue repair and remodeling. These findings were consistent with previous observations, detailing longitudinal changes of synovial mRNA expression of M1 and M2 markers 2–4 weeks after hemarthrosis in FVIII-KO mice [12]. Our observations here suggest that the type of FVIII preparation can influence this process based on the following observations. First, only mFcFVIII, but not rhFVIII, affected the ECM and collagen degradation pathways. Second, mFcFVIII reversed all highest DEGs, whereby most are known for their involvement in tissue repair and remodeling. Third, during the healing/transition phase, differential gene expression similarities between mIgG2a and mFcFVIII

could be identified. Forth, cell deconvolution analyses demonstrated that mFcFVIII enhanced the synovial macrophage population, and M1 and M2 marker expression profiling revealed different regulation patterns between mFcFVIII and rhFVIII. Together it appeared that mFcFVIII distinctly influences synovial reparative processes at later stages after hemarthrosis, different from rhFVIII.

Finally, it should be mentioned that both preparations also exerted unique off-target effects during the acute and healing phases. This was somewhat surprising, suggesting that different FVIII preparations may exert effects that are unrelated to injury/bleed. While these effects were not pronounced, they mainly involved immune regulatory pathways, whereby rhFVIII perturbed the immune system pathway on day 3, while mFcFVIII did not. Both preparations perturbed this pathway on day 14, albeit affecting different genes within the pathway.

We acknowledge that this study has several limitations. For instance, the directionality of gene expression changes and pathway perturbations in the context of FVIII reversal cannot be linked with certainty to overall positive or negative biological outcomes. Based on the favorable therapeutic effects of IVIG, [38] it is assumed though that Fc-mediated effects are overall beneficial. This is also supported by previous observations in relation to favorable hemophilic joint outcomes and rapid immune tolerization in patients [30–33,35–41], as well as M2 polarization and mitigation of immunogenicity in hemophilic mouse models [12]. Also, it is unclear to what extent the observed molecular effects, particularly given Fc species specificity, are applicable to human disease. Lastly, this study employed an acute bleed and treatment model, without claim to inform about effects on long-term synovial or joint health in the setting of FVIII prophylaxis.

Notwithstanding the limitation of direct linkage of transcriptomic changes to protein expression, the histological results demonstrating reduced synovial vascular remodeling and improved cartilage health with mFcFVIII provide at least some evidence that Fc-linkage for half-life extension may wield functions related to tissue repair. We, therefore, believe to provide proof-of-principle that the type of FVIII preparation administered acutely to control hemarthrosis can influence synovial processes beyond acute hemostasis control. These observations provide incentive to study potential beneficial effects of FVIII-Fc fusion protein for acute and chronic bleed control on clinical joint health outcomes in hemophilia.

## Supporting information

**S1 Fig. FVIII-KO hemarthrosis model.** Hemarthrosis was induced in FVIII-KO mice by sub-patellar needle puncture. Saline, rhFVIII, or mFcFVIII prophylaxis was given 2 h before and 6 h after injury (n = 3 to 5 per group). The extent of intra/peri-articular bleeding was determined by hematocrit measurement on day 2 after injury. Synovial tissue and blood were harvested at baseline, day 3, and day 14 post-injury. RNA was purified and analyzed by RNA sequencing using an Illumina NextSeq500 platform (75 bp; single-end). The R BioConductor packages tximport, edgeR, and limma were used to estimate counts from RSEM, trimmed mean of M-values (TMM) normalization was applied, and the limma-voom method was used for differential expression analyses (criteria: adjusted p-value <0.05). FVIII-KO, Factor VIII knock-out; mFcFVIII, mouse-specific Fc-fusion FVIII; rhFVIII, recombinant human FVIII; Bl, baseline.
(TIF)

**S2 Fig. Day 2 Hematocrit after induced hemarthrosis.** Hemarthrosis was induced by sub-patellar knee injury in FVIIIKO mice. Mice were treated with either saline (control) recombinant human factor VIII (rhFVIII/rhF8), mouse-specific Fc fusion factor VIII (mFcFVIII/rhF8), or mouse immunoglobulin (mIgG) 2a 2 hours before and 6 hours after the knee injury. Hematocrit was determined at baseline (Bl) mice and on day 2 after injury (n = 6–12 per group). Error bars represent the median with interquartile range values (***p < 0.001, ****p < 0.0001).
(TIF)

**S3 Fig. Effect of hemarthrosis and FVIII treatment on genes contributing to enriched pathways on day 3.** On-target and off-target subsets of DEGs (Figure 2) were subjected to enrichment analysis using gprofiler to identify

significant pathways from KEGG and Reactome databases. Representative pathways in each of the subsets are shown. Enriched pathways of group 1 was RNA processing (A). In group 2, the cytokine-cytokine interaction pathway was enhanced (B), in group 3, pathways related to immune system, RNA processing and metabolism were enriched (C). For group 4, metabolism and immune system pathways were enriched (D), and for group 6 cell adhesion and metabolism pathways were enriched (E). Extracellular matrix organization and cell cycle pathway were enriched in group 7 (F and G). DEGs, differentially expressed genes; KEGG, Kyoto Encyclopedia of Genes and Genomes; RNA, ribonucleic acid.

(TIF)

**S4 Fig. Visualization of gene expression +/− FVIII treatment after inducing hemarthrosis in synovium on day 3.** Heatmaps were generated for all the genes in groups 1 and 2 using the R function heatmap.2 with row normalization (Z-score). Each column represents gene expression from an individual mouse in each group: uninjured (baseline), injured-saline, and injured-rhFVIII treated (A), uninjured (baseline), injured-saline, and injured-mFcFVIII treated (B). rhFVIII, recombinant human factor VIII; mFcFVIII, mouse-specific Fc fusion factor VIII.

(TIF)

**S5 Fig. Effect of hemarthrosis and FVIII treatment on genes contributing to enriched pathways on day 14.** On-target and off-target subsets of DEGs were subjected to enrichment analysis using gprofiler to identify significant pathways from KEGG and Reactome databases. Representative pathways in each of the subsets are shown. Enriched pathways of group 2 were extracellular matrix organization and collagen degradation (A). In group 3 extracellular matrix organization, collagen degradation, and focal adhesion (B) were enriched. Groups 4 and 5 had enriched pathways for the immune system (C and D). Group 7 had enriched pathways predominantly relating to extracellular matrix organization (E) and metabolism (F). DEGs, differentially expressed genes; KEGG, Kyoto Encyclopedia of Genes and Genomes; ECM, extracellular matrix.

(TIF)

**S6 Fig. Visualization of synovial gene expression +/− FVIII treatment 14 days after injury/bleed.** Heatmaps were generated for all the genes in groups 1 and 2 using Rstudio with row normalization (Z-score). Each column represents gene expression from an individual mouse in each group: uninjured (baseline), injured-saline, and injured-rhFVIII treated (A), uninjured (baseline), injured-saline, and injured-mFcFVIII treated (B). rhFVIII, recombinant human factor VIII; mFcFVIII, mouse-specific Fc fusion factor VIII.

(TIF)

**S1 Table. Differentially expressed genes in synovial tissue on days 3 and 14.** Synovial tissue was harvested at baseline, 3- and 14-days post-injury/bleed with treatment by Saline (vehicle), rhFVIII, or mFcFVIII. Limma DGE analysis was performed (see Methods) between injured-saline vs baseline (day 3 or 14) samples, between injured-saline vs injured-rhFVIII, and injured-saline vs injured-mFcFVIII samples. The significant genes, with adjusted p-value <0.05, are listed for each comparison along with logFC and results from statistical analysis. DGE, differential gene expression; LogFC, Log-fold change.

(XLSX)

**S2 Table. All the pathways in synovium on days 3 and 14.** The DEGs were annotated using STRING interaction database. The resulting networks were analyzed using functional enrichment analysis which included KEGG and Reactome pathways with adjusted p-value (<0.05). Further visualization of this network was performed in Cytoscape. DEGs, differentially expressed genes; STRING, search tool for the retrieval of interacting genes; KEGG, Kyoto Encyclopedia of Genes and Genomes.

(XLSX)

**S3 Table. Effects of mIgG2a on DGEs in synovial tissue on days 3 and 14.** Synovial tissue was harvested at 3- and 14-days post-injury/bleed with treatment rhFVIII, mFcFVIII or mIgG2a. Limma DGE analysis was performed (see Methods) between injured-IgG2a vs injured-rhFVIII (day 3 or 14) and, between injured-mIgG2a vs injured-mFcFVIII. The significant genes, with adjusted p-value <0.05, are listed for each comparison along with logFC and results from statistical analysis. DGE, differential gene expression; LogFC, Log-fold change.
(XLSX)

**S4 Table. Pathways in synovium on days 3 and 14 (mIgG2a).** The DEGs were annotated using STRING interaction database. The resulting networks were analyzed using functional enrichment analysis which included KEGG and Reactome pathways with adjusted p-value (<0.05). DEGs, differentially expressed genes; STRING, search tool for the retrieval of interacting genes; KEGG, Kyoto Encyclopedia of Genes and Genomes.
(XLSX)

## Author contributions

**Conceptualization:** Annette von Drygalski.

**Data curation:** Bilgimol Chumappumkal Joseph, Thomas C Whisenant, Esther J Cooke.

**Formal analysis:** Bilgimol Chumappumkal Joseph, Thomas C Whisenant, Jenny Y Zhou, Nicca Falah, Annette von Drygalski.

**Funding acquisition:** Esther J Cooke, Annette von Drygalski.

**Investigation:** Annette von Drygalski.

**Methodology:** Esther J Cooke.

**Resources:** Annette von Drygalski.

**Software:** Thomas C Whisenant, Juan Andres De-Pablo Moreno.

**Supervision:** Annette von Drygalski.

**Writing – original draft:** Bilgimol Chumappumkal Joseph, Thomas C Whisenant, Annette von Drygalski.

**Writing – review & editing:** Bilgimol Chumappumkal Joseph, Thomas C Whisenant, Annette von Drygalski.

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
