## [Decision Letter · Decision Letter 0]

30 Dec 2024

PONE-D-24-47001Synovial Gene Expression after Hemarthrosis Differs Between FVIII-Deficient Mice Treated with Recombinant FVIII or FVIII-Fc Fusion ProteinPLOS ONE

Dear Dr. Bilgimol,

Thank you for submitting your manuscript to PLOS ONE. After careful consideration, we feel that it has merit but does not fully meet PLOS ONE’s publication criteria as it currently stands. Therefore, we invite you to submit a revised version of the manuscript that addresses the points raised during the review process.

We look forward to receiving your revised manuscript.

Kind regards,

Dominique Heymann, Ph.D.

Academic Editor

PLOS ONE

Journal Requirements:

"This study was funded by a Career Development Award from the National Hemophilia Foundation (NHF) (AvD), the NHF/Nicholas Cirelli Family Research Fund Judith Graham Pool Research Fellowship (EJC), by a research grant from Sanofi (AvD), and UL1TR001442 of CTSA (TCW)."

"This study was funded by a Career Development Award from the National Hemophilia Foundation (NHF) (AvD), the NHF/Nicholas Cirelli Family Research Fund Judith Graham Pool Research Fellowship (EJC), by a research grant from Sanofi (AvD), and UL1TR001442 of CTSA (TCW)."

"This study was funded by a Career Development Award from the National Hemophilia Foundation (NHF) (AvD), the NHF/Nicholas Cirelli Family Research Fund Judith Graham Pool Research Fellowship (EJC), by a research grant from Sanofi (AvD), and UL1TR001442 of CTSA (TCW)."

"I have read the journal's policy and the authors of this manuscript have the following competing interests:

AvD has received honoraria for participating in scientific advisory board panels, consulting, and speaking engagements for BioMarin, Pfizer, Bioverativ/Sanofi, CSL-Behring, Novo Nordisk, Spark Therapeutics, Takeda and Regeneron. AvD is co-founder of Hematherix LLC., a biotech company that is developing superFVa therapy for bleeding complications. AvD is a member of the Board of Directors of Hematherix LLC."

7. We note that Figure S1 in your submission contain copyrighted images. All PLOS content is published under the Creative Commons Attribution License (CC BY 4.0), which means that the manuscript, images, and Supporting Information files will be freely available online, and any third party is permitted to access, download, copy, distribute, and use these materials in any way, even commercially, with proper attribution. For more information, see our copyright guidelines: http://journals.plos.org/plosone/s/licenses-and-copyright.

a. You may seek permission from the original copyright holder of Figure S1 to publish the content specifically under the CC BY 4.0 license. 

Reviewers' comments:

Reviewer's Responses to Questions

**Comments to the Author**

1. Is the manuscript technically sound, and do the data support the conclusions?

Reviewer #1: Partly

Reviewer #2: Yes

2. Has the statistical analysis been performed appropriately and rigorously? 

Reviewer #1: Yes

Reviewer #2: Yes

3. Have the authors made all data underlying the findings in their manuscript fully available?

Reviewer #1: Yes

Reviewer #2: Yes

4. Is the manuscript presented in an intelligible fashion and written in standard English?

Reviewer #1: Yes

Reviewer #2: Yes

5. Review Comments to the Author

Reviewer #1: This study examines at day 3 and day 14, the effects of murine FVIII-Fc fusion protein (mFcFVIII) in comparison to recombinant human FVIII (rhFVIII) on synovial molecular processes in a context of hemarthrosis. Hemarthrosis were induced in FVII-KO mice by sub-patellar needle puncture and an approach bases on RNAseq and differential gene expression analysis complimented by KEGG and Reactome pathway analysis were applied to synovial tissue. The authors observe a small proportion of genes affected by FVIII treatment and distinct differences between both FVIII concentrates, both related and unrelated to injury/bleed. At day 3, mFcVIII had effects on immune and inflammatory processes whereas rhFVIII seems more modulated mRNA and protein processes. At day 14, molecular profiling revealed expression changes in favour of a transition to synovial tissue repair and remodelling and the authors argue that mFcFVIII compared to rhFVIII facilitated the M1 to M2 macrophage transition. The study, only based on transcriptomic analysis, seems methodologically well conducted, but some issues need to be addressed:

Major remarks:

- The limitations of studies based solely on transcriptomic approaches are not indicated in discussion. These limitations must be added.

- In results section, the authors written at the beginning of the section “injured mice treated with rhFVII vs mFcFVII revealing similar expression profiles without statistical difference” that is in contradiction with the results that follow. Can the authors revise the wording of this sentence to be more explicit?

- The classification of genes/pathways as on-target (related to injury/bleed) and off-target (unrelated to injury/bleed) is not very clear. For example, in results, cytokine-cytokine receptor interaction is on-target and perturbation of immune system is off-target and focal adhesion is on-target and cell adhesion is off-target. This point needs to be clarified.

- In results section, the second paragraph focus on used methodology and should be moved in material and methods section.

- Did the authors apply the cell-type deconvolution from bulk RNASeq using immune cell references at day 3. It would be interesting to present this data.

- Concerning the differential expression levels of macrophage subtype-specific markers M1 (figure 5B), the authors indicated in results section “among the 5 M1 markers, only il6 was mildly increased after saline treatment compared to baseline and decreased with both FVIII-preparations, indicating overall abating inflammation”. The effect observed is very modest and the authors seem to have overinterpreted their results.

- It is the same remark for the discussion about these results (“mFcFVIII facilitate a polarization to M2, not observed to the same extent with rhFVIII)”. Concerning the M2 markers, the difference between rhFVIII and mFcFVIII for lrf4 seems minimal. The last sentence of the results needs to be modified. What are the results for the anti-inflammatory M2 macrophages markers?

Minor remarks:

- The use of abbreviations needs to be reviewed. There should be no abbreviation in the abstract and once an abbreviation is defined in the text it should then be systematically used.

- The title of the Table I is cut.

- In supplementary figures, the figure 1 is illegible for the part on the right.

Reviewer #2: The authors used next-generation RNA sequencing (RNA-Seq) to analyse the temporal evolution of molecular processes in synovial tissue after haemarthrosis in a mouse model of haemophilia A. They studied changes in gene expression and disruption of molecular pathways with or without treatment with factor VIII (FVIII) on days 3 and 14 after haemarthrosis. In addition, they compared the effects of murine FVIII fused to an Fc portion (mFcFVIII) with those of recombinant human FVIII (rhFVIII, without Fc portion) to determine whether this Fc portion influences processes linked to inflammation, tissue repair or immune responses. The results show that in the acute phase (D3), mFcFVIII had unique on-target effects related to immune and inflammatory regulation, whereas rhFVIII primarily affected mRNA and protein processing. At day 14, macrophage profiling indicated a transition from M1 to M2, and only mFcFVIII uniquely influenced pathways and genes associated with tissue remodelling and repair.

This study, conducted by a team with internationally recognized expertise in the field and supported by prior publications on the models used, is robustly designed and executed. The limitations of the work are clearly and transparently presented. The most innovative findings stem from the use of mFcFVIII, which, after 14 days, demonstrated a unique influence on molecular pathways and genes associated with tissue remodeling and repair. The authors attribute this beneficial effect to the mFc portion of the molecule.

However, this conclusion could be further strengthened by including one or two control groups of hemophilia A mice treated either with isolated mFc or with free rhFVIII combined with mFc. Additionally, the study would benefit from experimental evidence, such as histological observations, explicitly demonstrating the beneficial effects of mFcFVIII on vascular tissues in hemophilic mice.

6. PLOS authors have the option to publish the peer review history of their article (what does this mean? ). If published, this will include your full peer review and any attached files.

**Do you want your identity to be public for this peer review?** For information about this choice, including consent withdrawal, please see our Privacy Policy .

Reviewer #1: No

Reviewer #2: No

---

## [Author Response · Author response to Decision Letter 1]

27 Jan 2025

Manuscript Number: PONE-D-24-47001

Synovial Gene Expression after Hemarthrosis Differs Between FVIII-Deficient Mice Treated with Recombinant FVIII or FVIII-Fc Fusion Protein

Comment 1

Answer 1

Thank you for the formatting instructions. We have updated the manuscript to meet PLOS ONE’s style requirements

Comment 2

To comply with PLOS ONE submissions requirements, in your Methods section, please provide additional information regarding the experiments involving animals and ensure you have included details on (1) methods of sacrifice, (2) methods of anesthesia and/or analgesia, and (3) efforts to alleviate suffering.

Answer 2

Thank you

We have added additional information to the method section on pages 7 and 8 regarding anesthesia, efforts to alleviate suffering, and methods of sacrifice

Comment 3

Thank you for stating the following financial disclosure:

"This study was funded by a Career Development Award from the National Hemophilia Foundation (NHF) (AvD), the NHF/Nicholas Cirelli Family Research Fund Judith Graham Pool Research Fellowship (EJC), by a research grant from Sanofi (AvD), and UL1TR001442 of CTSA (TCW)."

Answer 3

Thank you for the comment. We have revised the financial disclosure as suggested.

Financial Disclosure now reads as follows:

“Financial Disclosure

This study was funded by the National Hemophilia Foundation/Nicholas Cirelli Family Research Fund Judith Graham Pool Research Fellowship (EJC), by a research grant from Sanofi (AvD), and UL1TR001442 of CTSA (TCW). The funders had no role in study design, data collection, and analysis. Sanofi reviewed and provided feedback on the manuscript. The authors had full editorial control of the manuscript and provided their final approval of all content.’’.

Comment 4

Thank you for stating the following in the Acknowledgments Section of your manuscript:

"This study was funded by a Career Development Award from the National Hemophilia Foundation (NHF) (AvD), the NHF/Nicholas Cirelli Family Research Fund Judith Graham Pool Research Fellowship (EJC), by a research grant from Sanofi (AvD), and UL1TR001442 of CTSA (TCW)."

"This study was funded by a Career Development Award from the National Hemophilia Foundation (NHF) (AvD), the NHF/Nicholas Cirelli Family Research Fund Judith Graham Pool Research Fellowship (EJC), by a research grant from Sanofi (AvD), and UL1TR001442 of CTSA (TCW)."

Answer 4

Thank you, we eliminated the Acknowledgement Section and added a Financial Disclosure.

Comment 5

Thank you for stating the following in the Competing Interests section:

"I have read the journal's policy and the authors of this manuscript have the following competing interests:

AvD has received honoraria for participating in scientific advisory board panels, consulting, and speaking engagements for BioMarin, Pfizer, Bioverativ/Sanofi, CSL-Behring, Novo Nordisk, Spark Therapeutics, Takeda and Regeneron. AvD is co-founder of Hematherix LLC., a biotech company that is developing superFVa therapy for bleeding complications. AvD is a member of the Board of Directors of Hematherix LLC."

Answer 5

We confirm that this does not alter our adherence to PLOS ONE policies.

Comment 6

Please include your full ethics statement in the ‘Methods’ section of your manuscript file. In your statement, please include the full name of the IRB or ethics committee who approved or waived your study, as well as whether or not you obtained informed written or verbal consent. If consent was waived for your study, please include this information in your statement as well.

Answer 6

Thank you,

All the protocols were approved by the University of California San Diego Institutional Animal Care and Use Committee (IACUC). The information is included in the method section on page 7

Comment 7

We note that Figure S1 in your submission contain copyrighted images. All PLOS content is published under the Creative Commons Attribution License (CC BY 4.0), which means that the manuscript, images, and Supporting Information files will be freely available online, and any third party is permitted to access, download, copy, distribute, and use these materials in any way, even commercially, with proper attribution. For more information, see our copyright guidelines: http://journals.plos.org/plosone/s/licenses-and-copyright.

a. You may seek permission from the original copyright holder of Figure S1 to publish the content specifically under the CC BY 4.0 license.

Answer 7

Thank you for the comment, we have obtained the required license from Biorender for Figure S1 and the same is uploaded into the portal

Comment 8

Please include captions for your Supporting Information files at the end of your manuscript, and update any in-text citations to match accordingly. Please see our Supporting Information guidelines for more information: http://journals.plos.org/plosone/s/supporting-information

Answer 8

Accomplished as instructed

Review Comments to the Author

Reviewer #1:

This study examines at day 3 and day 14, the effects of murine FVIII-Fc fusion protein (mFcFVIII) in comparison to recombinant human FVIII (rhFVIII) on synovial molecular processes in a context of hemarthrosis. Hemarthrosis were induced in FVII-KO mice by sub-patellar needle puncture and an approach bases on RNAseq and differential gene expression analysis complimented by KEGG and Reactome pathway analysis were applied to synovial tissue. The authors observe a small proportion of genes affected by FVIII treatment and distinct differences between both FVIII concentrates, both related and unrelated to injury/bleed. At day 3, mFcVIII had effects on immune and inflammatory processes whereas rhFVIII seems more modulated mRNA and protein processes. At day 14, molecular profiling revealed expression changes in favour of a transition to synovial tissue repair and remodelling and the authors argue that mFcFVIII compared to rhFVIII facilitated the M1 to M2 macrophage transition. The study, only based on transcriptomic analysis, seems methodologically well conducted, but some issues need to be addressed:

Major remarks:

Comment 1

The limitations of studies based solely on transcriptomic approaches are not indicated in discussion. These limitations must be added.

Answer 1

Thank you for this comment. We agree with the Reviewer (also see Comment 2 of Reviewer 2). We, therefore, examined joint histology sections to determine if there were direct effects on tissue repair between the two FVIII preparations. We did find that mFcFVIII had positive effects on the reduction of synovial vascular remodelling and cartilage health compared to rhFVIII. We added a paragraph to the Result Section named “ Synovial and cartilage changes after hemarthrosis” (page 20).

To acknowledge the limitations of transcriptomic analyses, but also introduce our preliminary evidence of confirmation of transcriptomic findings on the tissue level we changed the last paragraph of the Discussion as follows:

“Notwithstanding the limitation of direct linkage of transcriptomic changes to protein expression the histology results related to synovial vascular remodeling and cartilage health provide at least some evidence that Fc-linkage for half-life extension of FVIII may wield functions related to tissue repair. We, therefore, believe to provide proof-of-principle that the type of FVIII preparation administered acutely to control hemarthrosis can influence synovial processes beyond acute hemostasis control. These observations provide incentive to study potential beneficial effects of FVIII-Fc fusion protein for acute and chronic bleed control on clinical joint health outcomes in hemophilia.”

Comment 2

In results section, the authors written at the beginning of the section “injured mice treated with rhFVII vs mFcFVII revealing similar expression profiles without statistical difference” that is in contradiction with the results that follow. Can the authors revise the wording of this sentence to be more explicit?

Answer 2

Thank you for the suggestion, we have removed that sentence since it is indeed confusing and moved the remainder of the paragraph to the Method section (also see Comment/Answer 4).

Comment 3

The classification of genes/pathways as on-target (related to injury/bleed) and off-target (unrelated to injury/bleed) is not very clear. For example, in results, cytokine-cytokine receptor interaction is on-target and perturbation of immune system is off-target and focal adhesion is on-target and cell adhesion is off-target. This point needs to be clarified.

Answer 3

Thank you for the comment.

We have added a detailed description of “on-target” and “off-target” genes/effects to the Method section (Pages 9-10). To this end, we also re-labelled supplementary table 2 (S2 Table), which lists the pathways falling into the on- and off-target categories.

Comment 4

In results section, the second paragraph focus on used methodology and should be moved in material and methods section.

Answer 4

Thank you for the suggestion.

We moved the second paragraph to the Methods section, which now includes improved explanations for on- and off-target effects as well as group comparisons. This paragraph now reads as follows:

“Gene expression analysis after hemarthrosis

Gene expression profiles related to FVIII treatments were defined as "on-target" (related to injury/bleed), in contrast to "off-target" (unrelated to injury/bleed) effects. Both, “on-target” and “off-target” genes are differentially expressed between injured-saline vs. injured-rhFVIII/mFcFVIII treatments; however, only “on-target” genes are differentially expressed between injured-saline vs. baseline. Synovial gene expression was compared as follows on days 3 and 14: i) “Baseline” (uninjured FVIII-KO mice) vs. FVIII-KO injured mice treated with saline (“injured-saline”); ii) injured-saline vs. FVIII-KO injured mice treated with rhFVIII (“injured-rhFVIII”); iii) injured-saline vs. FVIII-KO injured mice treated with mFcFVIII (“injured-mFcFVIII”) (n=3-5 mice per group, Supplementary Fig 1). DGE changes related to injury/bleed were identified by comparing baseline to injured-saline mice. To elucidate the treatment effects of each FVIII-preparation, injured-saline were compared to injured-FVIII treated mice (injured-rhFVIII or injured-mFcFVIII on days 3 and 14).’’

Comment 5

Did the authors apply the cell-type deconvolution from bulk RNASeq using immune cell references at day 3. It would be interesting to present this data.

Answer 5

Thank you for the suggestion. We included these analyses, and added them to the Result section “Cell-type deconvolution from bulk RNASeq using immune cell references” and adjusted the Discussion section accordingly (paragraph 4 of discussion).

Comment 6

Concerning the differential expression levels of macrophage subtype-specific markers M1 (figure 5B), the authors indicated in results section “among the 5 M1 markers, only il6 was mildly increased after saline treatment compared to baseline and decreased with both FVIII-preparations, indicating overall abating inflammation”. The effect observed is very modest and the authors seem to have overinterpreted their results.

Answer 6

Thank you for the comment. We agree with the reviewer and softened the description of the results (Page 19).

Comment 7

It is the same remark for the discussion about these results (“mFcFVIII facilitate a polarization to M2, not observed to the same extent with rhFVIII)”. Concerning the M2 markers, the difference between rhFVIII and mFcFVIII for lrf4 seems minimal. The last sentence of the results needs to be modified. What are the results for the anti-inflammatory M2 macrophages markers?

Answer 7

Thank you for the comment.

This ties in with the responses to comments 5 and 6. Due to the inclusion of day 3 deconvolution analyses we have rewritten and adjusted the appropriate Result and Discussion Sections (see comments 5 and 6). In general, we hope to have clarified the role of transcriptomic

---

## [Decision Letter · Decision Letter 1]

18 Feb 2025

Synovial Gene Expression after Hemarthrosis Differs Between FVIII-Deficient Mice Treated with Recombinant FVIII or FVIII-Fc Fusion Protein

PONE-D-24-47001R1

Dear Dr. Joseph,

We’re pleased to inform you that your manuscript has been judged scientifically suitable for publication and will be formally accepted for publication once it meets all outstanding technical requirements.

Kind regards,

Dominique Heymann, Ph.D.

Academic Editor

PLOS ONE

Additional Editor Comments (optional):

Reviewers' comments:

Reviewer's Responses to Questions

**Comments to the Author**

1. If the authors have adequately addressed your comments raised in a previous round of review and you feel that this manuscript is now acceptable for publication, you may indicate that here to bypass the “Comments to the Author” section, enter your conflict of interest statement in the “Confidential to Editor” section, and submit your "Accept" recommendation.

Reviewer #1: All comments have been addressed

Reviewer #2: All comments have been addressed

2. Is the manuscript technically sound, and do the data support the conclusions?

Reviewer #1: Yes

Reviewer #2: Yes

3. Has the statistical analysis been performed appropriately and rigorously? 

Reviewer #1: Yes

Reviewer #2: Yes

4. Have the authors made all data underlying the findings in their manuscript fully available?

Reviewer #1: Yes

Reviewer #2: Yes

5. Is the manuscript presented in an intelligible fashion and written in standard English?

Reviewer #1: Yes

Reviewer #2: Yes

6. Review Comments to the Author

Reviewer #1: (No Response)

Reviewer #2: All my comments have been thoroughly addressed by the authors. The additional experiments have been properly conducted, and the text has been revised accordingly.

7. PLOS authors have the option to publish the peer review history of their article (what does this mean? ). If published, this will include your full peer review and any attached files.

**Do you want your identity to be public for this peer review?** For information about this choice, including consent withdrawal, please see our Privacy Policy .

Reviewer #1: **Yes: ** Annabelle Dupont, Université de Lille, CHU de Lille service d'hémostase et transfusion, UMR Inserm 1011

Reviewer #2: No

---

## [Editor Report · Acceptance letter]

PONE-D-24-47001R1

PLOS ONE

Dear Dr. Chumappumkal Joseph,

I'm pleased to inform you that your manuscript has been deemed suitable for publication in PLOS ONE. Congratulations! Your manuscript is now being handed over to our production team.

Kind regards,

on behalf of

Pr. Dominique Heymann

Academic Editor

PLOS ONE